# Metabolites Produced by a New *Lactiplantibacillus plantarum* Strain BF1-13 Isolated from Deep Seawater of Izu-Akazawa Protect the Intestinal Epithelial Barrier from the Dysfunction Induced by Hydrogen Peroxide

**DOI:** 10.3390/md20020087

**Published:** 2022-01-20

**Authors:** Xiaozhen Diao, Katsuhisa Yamada, Yuji Shibata, Chiaki Imada

**Affiliations:** 1Applied Microbiology Lab, Course of Applied Marine Biosciences, Graduate School of Marine Science and Technology, Tokyo University of Marine Science and Technology, Tokyo 108-8477, Japan; k-yamada@dhc.co.jp (K.Y.); imada@kaiyodai.ac.jp (C.I.); 2DSW Laboratory of DHC Co., Ltd., Tokyo 106-0047, Japan; yshibata@dhc.co.jp

**Keywords:** *Lactiplantibacillus* *plantarum*, deep seawater, tight junctions, aquaporin3, hydrogen peroxide, lactic acid

## Abstract

This study aimed to investigate the protective effect of the metabolites produced by a new *Lactiplantibacillus plantarum* strain BF1-13, isolated from deep seawater (DSW), on the intestinal epithelial barrier against the dysfunction induced by hydrogen peroxide (H_2_O_2_) and to elucidate the mechanism underlying the effect. Protective effect of the metabolites by strain BF1-13 on the barrier function of the intestinal epithelial model treated with H_2_O_2_ was investigated by the transepithelial electrical resistance (TEER). The metabolites enhanced the Claudin-4 (CLDN-4) expression, including at the transcription level, indicated by immunofluorescence staining and quantitative RT-PCR. The metabolites also showed a suppression of aquaporin3 (AQP3) expression. Lactic acid (LA) produced by this strain of homofermentative lactic acid bacteria (LAB) had a similar enhancement on CLDN-4 expression. The metabolites of *L. plantarum* strain BF1-13 alleviated the dysfunction of intestinal epithelial barrier owing to its enhancement on the tight junctions (TJs) by LA, along with its suppression on AQP3-facilitating H_2_O_2_ intracellular invasion into Caco-2 cells. This is the first report on the enhancement of TJs by LA produced by LAB.

## 1. Introduction

It is known that the intestinal epithelium basically functions for the maintenance of the internal environment [1]. It absorbs nutrients and defends the exogenous pathogens and their secretion of toxins to protect the human body from diseases, including inflammatory bowel disease (IBD), celiac diseases, and diarrhea [2]. Between the intestinal epithelial cells, there exist TJs mediating the paracellular permeability to achieve the barrier function.

TJs are complex, contiguous, relatively, and dynamically impermeable junctional devices that take up the most apical surface of the intercellular space of intestinal epithelium [3]. TJs consist of several functional proteins including claudins, occludin (OCLN), tricellulin, and junctional adhesion molecules. The barrier function of the intestinal epithelium is closely related to these functional proteins’ expression [4]. TEER assay with the intestinal epithelial model constructed by Caco-2 cells has been widely used to investigate the integrity of the intestinal epithelial barrier [5].

However, the function of TJs can be degraded by some reactive oxygen species, such as H_2_O_2_ [6]. H_2_O_2_ can be generated from the endogenous, such as the antibacterial defense by immune response or from the exogenous, such as unhealthy lifestyle, environmental pollution, and aging. Although the H_2_O_2_ at a low level can participate in cell signaling and cell proliferation and can eliminate invading pathogens, the high concentration may induce oxidative stress-related diseases, such as diabetes and inflammatory bowel disease in the case of the gastrointestinal tract [7,8]. Oxidative stress induced by different kinds of reactive oxygen species has been proven to cause the disruption of intestinal epithelial TJs [9]. It has been recently reported that the transportation of endogenous H_2_O_2_ facilitated by peroxiporins such as AQP3 (transcellular invasion) occurs prior to simple diffusion (paracellular invasion) [10].

The gut microbiota maintains the integrity and function of the gastrointestinal tract, where there are colonies of the huge community of LA-producing bacteria such as LABs [11]. LA was found as an impure component from sour milk in 1780, which was proven to be an isolated metabolite of microorganisms by Pasteur in 1857. In 1881, a French scientist successfully produced it by microbial fermentation, which leads to the first boom of LA industrial production. Since then, LA has been continuously among the hot spots in many industries, such as the food, probiotics, medical, cosmetic, and pharmaceutical industry. From the 2000s, its biotechnological application to form polylactic acid (PLA) has been of increasing interest [12]. Due to its advantage with lower cost, less limitation on production, high efficiency, and being eco-friendly, bacterial fermentation takes up about 90% of the industry market of LA production [13]. So far, there is no report on the effect of LA on the intestinal epithelial barrier function, although there are many about the LAB themselves.

Grishina et al. [14] proved that kefir and ayran supernatants, which contain large amounts of LA, decreased DNA damage in colon cells caused by fecal water in vitro. However, according to them, no effect on intestinal TJs was observed. Kefir & ayran (fermented milk) fermented by complicated bacterial species produce various components in the culture supernatants, including not only lactic acid but also other organic substances, such as ethanol [15]. Due to all these reasons, their results cannot be convincing evidence for the effect of LA on TJs. In contrast, Forsyth G. W. et al. [16] reported that L-LA reduced the net fluid secretion in ligated jejunal loops in vitro caused by cholera toxin, which indicated the potential effect of LA on intestinal epithelial permeability regulated by TJs.

It has been proven that the intestinal oxidative damage induced by H_2_O_2_ can be inhibited or released by some species of *Lactobacillus*, mostly isolated from the probiotics, [17,18] or by the fermentative products of them [19]. Although homofermentative LAB mainly refers to both *L. plantarum* and *Lactobacillus delbrueckii*, unlike the former, *L. delbrueckii*, which can convert 95% of glucose into LA, only produces D-LA [16], which was reported to be harmful by inducing acidosis and de-calcification [20]. The *L. plantarum* strain JCM11125 (also as *L. arizonensis* strain; 1474 bp; ENA accession number AJ965482) isolated from jojoba meal fermentation is a homofermentative strain with no catalase production and had a wide range of growth temperatures [21], which was used as the standard strain in this study.

This study aims to investigate the protective effect of the metabolites produced by DSW-derived *L. plantarum* strain BF1-13 on the intestinal epithelial barrier against the H_2_O_2_-induced dysfunction and its underlying mechanism related to the AQP-facilitating H_2_O_2_ intracellular invasion. Furthermore, it provides the potential application of the strain BF1-13 fermented food and its metabolites as supplements for IBD patients.

## 2. Results

### 2.1. Characteristics of Three Lactiplantibacillus Strains

The two isolated *L. plantarum* strains were confirmed by 16S rRNA gene sequencing and named as BF1-13 (1492 bp; DDBJ accession number LC666821) and H-6 (1492 bp; DDBJ accession number LC666820), respectively.

These three strains showed nearly the same characteristics. There is no difference in salt tolerance among the strains, whose maximum tolerances were 5% NaCl. Similar to the standard strain JCM11125, both of the two isolated strains were able to ferment all kinds of substrates and survived in the nutrients-limited medium only supplemented with specific carbohydrates All strains grew at a pH range of 2–9. However, strain BF1-13 failed to grow at 20 °C and showed a preference for a higher temperature (27–40 °C). (Table 1)

During the 12 h incubation (pH 6.5) at 37 °C, strain BF1-13 showed superiority in cell growth with the other two strains. Among the three strains, strain BF1-13 showed the fastest growth (1.67 × 10^9^ CFU/mL, Figure 1a), and the most remarkable decreasing pH value (pH 4.599) (Figure 1b). From the measurement of the concentration of LA contained in each culture supernatant (CS), it was suggested that there was a little difference in the LA production during 12 h incubation in the following groups: strain BF1-13 (26.3 mM), strain H-6 (20.0 mM), the standard strain JCM11125 (19.8 mM) (Table 2). As mentioned above, strain BF1-13 was selected as the research target because it showed superiority in growth and LA accumulation. Strain JCM 1112 was prepared as the standard strain in this study.

### 2.2. Protection on the Intestinal Epithelial Barrier by the Metabolites Produced by Strain BF1-13 against the Dysfunction Caused by H_2_O_2_ Treatment in Caco-2 Cells

TcTEER values decreased to 45.1% of the initial values after 6 h H_2_O_2_ treatment on the intestinal epithelial model using Caco-2 cells. The decreasing tcTEER value by H_2_O_2_ treatment was suppressed by the supplementation of 5% (*v/v*) CS of strain BF1-13. At the same supplemented concentration of 5% (*v/v*) of the CSs, strain BF1-13 (62.2%) showed slightly higher TEER values than standard strain (59.9%) (Figure 2b). However, this effect of the CS of strain BF1-13 disappeared by the 10% (*v/v*) supplementation (28.3%). (Figure 2a).

### 2.3. Enhancement on the TJs-Related Proteins by the Metabolites Produced by Strain BF1-13 from the Suppression Induced by H_2_O_2_

#### 2.3.1. Enhancement on CLDN-4 by the Metabolites Produced by Strain BF1-13 through Immunofluorescent Microscopy

Decreasing expression of CLDN-4 by H_2_O_2_ treatment was inhibited by the supplementation of the CS of the strain BF1-13. This effect was also shown on the standard strain (Figure 3). However, the expression of OCLN was different from that of CLDN-4. There was a little difference between the metabolites of the two strains. In the case of strain BF1-13, the difference of the effects between supplemented concentration 5% and 10% was barely seen with the immunofluorescent staining microscopy (Data not shown.)

#### 2.3.2. Inducement on CLDN-4 mRNA by the Metabolites Produced by Strain BF1-13 through Quantitative RT-PCR

The mRNA expression of CLDN-4 was only induced by the supplementation with the CS of the strain BF1-13 in Caco-2 cells (Figure 4a). The mRNA expression of CLDN-4 with the supplementation with the CS of strain BF1-13 (fold-change of 1.75 & 1.53) was higher than that with the NC (fold-change of 1.20 & 1.15) during the 2 h H_2_O_2_ treatment. However, enhancement of CLDN-4 expression was not shown on that treatment with the CS of the standard strain (fold-change of 0.85 & 0.91). In the case of OCLN, the mRNA expression was not maintained by the supplementations with the CSs of both two strains (Figure 4b).

### 2.4. Suppression on AQP3 by the Metabolites Produced by Strain BF1-13

#### 2.4.1. Suppression on AQP3 by the Metabolites Produced by Strain BF1-13 through Immunofluorescent Microscopy

The expression of AQP3 was enhanced by 1 mM H_2_O_2_ treatment compared to the blank, which was also supported by the previous report [10]. The AQP3 expression was suppressed by the supplementation with the CS of strain BF1-13. Then, the same effect was shown in the CS of the standard strain (Figure 5).

#### 2.4.2. Suppression on AQP3 mRNA by the Metabolites Produced by Strain BF1-13 through Quantitative RT-PCR

The mRNA expression of AQP3 was significantly suppressed by the metabolites by strain BF1-13. This suppression was also shown on the metabolites by the standard strain (Figure 6). The mRNA expression of AQP3 with the supplementation with the CS of strain BF1-13 (fold-change of 0.57 & 0.52) was strongly suppressed during the 2 h H_2_O_2_ treatment, as well as that of the standard strain (fold-change of 0.45 & 0.45).

### 2.5. Effect by LA Contained in the Metabolites Produced by Strain BF1-13 on the Intestinal Epithelial Barrier with H_2_O_2_ Treatment

#### 2.5.1. Protection on the Intestinal Epithelial Barrier by LA against the Dysfunction Caused by H_2_O_2_ Treatment

Equal concentration of authentic LA (1.32 mM), which was contained in the metabolites, included in the 5% (*v/v*) CS of strain BF1-13 indicated almost the same protective effect on decreasing TEER by H_2_O_2_ treatment on the model (Figure 7a). Supplementation of authentic LA (1.32, 1.97, and 2.63 mM) suggested the significant maintenance effect of relative TEER values (83%, 74%, 74%) on the model treated with H_2_O_2_ (Figure 7b). However, the protective effect of LA decreased by the higher concentration, which was more than 1.32 mM.

#### 2.5.2. Enhancement on TJs-Related Proteins by LA Contained in the Metabolites Produced by Strain BF1-13

The CLDN-4 expression was maintained by the supplementation of the CS of strain BF1-13 and the equal concentration of authentic LA against H_2_O_2_ treatment (Figure 8). There was little difference in the effect between them. However, in the case of the equal concentration of authentic LA contained in the metabolites, the suppression of OCLN expression by the H_2_O_2_ treatment was relieved.

The mRNA expression of CLDN-4 was induced by the supplementation with the CS of strain BF1-13. This effect was also shown in the equal concentration of authentic LA (fold-change of 1.55 & 1.26) (Figure 9a). However, the mRNA expression of OCLN was only induced by the supplementation with LA at 0.5 h (fold-change of 1.25) (Figure 9b).

#### 2.5.3. Effect on AQP3 by LA Contained in the Metabolites Produced by Strain BF1-13

The LA (1.32 mM) contained in the metabolites of strain BF1-13 showed no suppressive effect on AQP3 expression (Figure 10) nor on its mRNA expression (Figure 11).

## 3. Discussion

The *L. plantarum* strain BF1-13 grew at a wide range of temperature (27–40 °C) and pH (2–9), which also showed good tolerance to salt (up to 5% NaCl). The optimal culture conditions for *L. plantarum* strain BF1-13 were determined as the anaerobic incubation around pH 6.5 at 37 °C. The LA contained in the CS after 12 h incubation of strain JCM11125 was less than that of strain BF1-13, which is consistent with the fact it can only convert 80% of the glucose into LA [22]. Good tolerance to low pH allows the strain BF1-13 to colonize in an acidic environment such as the human intestine. Its utilizing of the single carbohydrates source overcomes food intolerances during consumption, such as lactose intolerance. In summary, strain BF1-13 was suggested to have less limitation on both production and consumption when applied to different industries.

When the mechanism on the maintenance of human health via oral intake by the metabolites produced by marine-derived strain is studied, it is more practical to combine it with the transportation routes through the epithelial membrane during their absorptions in the small intestine. The small intestine is the main place for the nutrients to be taken up from the intestinal lumen to reach the lamina propria through the intestinal epithelium. In this study, all the tested samples were supplemented from the apical side, which stands for the lumen in vivo. As an endogenous reactive oxygen species, H_2_O_2_ is mostly produced in the lamina propria by antibacterial defense. Therefore, the monolayers in vitro model constructed by Caco-2 cells, which imitated the intestinal epithelial barrier, was treated with H_2_O_2_ from the basolateral side in this study. As a result, it was suggested by the relative TEER value that the metabolites produced by *Lactiplantibacillus plantarum* strain BF1-13 protected the intestinal epithelial barrier from the dysfunction caused by H_2_O_2_ treatment. The equal amount of LA included in the metabolites of strain BF1-13 also showed the same protective effect on the barrier. According to the results shown by immunofluorescence staining and quantitative RT-PCR, LA (1.32 mM) contained in the CS of strain BF1-13 showed the similar enhancement on CLDN-4 expression at the transcription level with the metabolites themselves. In summary, it was suggested that LA, which was the only organic acid included in the metabolites produced by homofermentative LAB, was one of the bioactive substances to enhance the expression of TJs-related protein especially CLDN-4.

However, this enhancement on OCLN was much weaker than CLDN-4 especially by the supplementation of the CSs of two strains. This difference between CLDN-4 and OCLN was shown on both the protein expressions and the mRNA expression. To investigate why the protective effect on the barrier function of the intestinal epithelial model was achieved by the different enhancement on TJs expressions, further experiments related to the expression of AQP3, which was recently proven to facilitate H_2_O_2_ intracellular invasion, was prepared in this study [23]. It was known that AQP3 was expressed only on the basolateral side of the cell membrane [24]. Combining the positions of CLDN-4 and OCLN in TJs structure, it was indicated that H_2_O_2_ added to the basolateral side of the monolayer in the model had two different invasion routes, including intercellular invasion by simple passive diffusion and intracellular route by AQP3-facilitating diffusion. It was suggested that H_2_O_2_ intracellular invasion via APQ3 caused the suppression of the expression of TJs-related proteins, especially CLDN-4. It was also suggested that the suppression by the metabolites produced by the strain on AQP3 expression at the transcription level made the H_2_O_2_ preferentially target on OCLN via intercellular route than CLDN-4 via AQP3-facilitating intracellular invasion. The suppression was not shown by LA, mostly because, as a small molecule, it prefers the paracellular route by TJs rather than the transcellular route.

*Lactiplantibacillus plantarum* strain BF1-13 isolated from DSW in Izu-Akazawa was suggested to be a fast-growing strain with a higher LA accumulation compared to the same species strains isolated from surface seawater and terrestrial. According to this, strain BF1-13 was suggested to have the possibility of wide application in the LAB market. In addition, the protective effect of the metabolites produced by strain BF1-13 discovered by this study will provide the possibility of the application in the maintenance of human health against intestinal related diseases. Moreover, the relationship between the dysfunction of intestinal epithelium and H_2_O_2_ intracellular invasion has been revealed for the first time. This is the first time that LA has been reported to be an essential substance for the enhancement on the intestinal epithelial barrier from the dysfunction caused by H_2_O_2_, although many other effects of LA have been well-known. Still, the suppression of the AQP3 expression by the metabolites remains to be further researched.

## 4. Materials and Methods

### 4.1. Chemicals

High-glucose DMEM (HG-DMEM) was prepared with Dulbecco’s modified Eagle’s medium (DMEM; Nissui, Tokyo, Japan). De-Man Rogosa Sharpe medium (MRS broth, BD Difco^TM^, Sparks, MD, USA) was purchased for the incubation of LAB. Rabbit anti-CLDN-4, OCLN, AQP3 antibodies and horseradish peroxidase conjugated Goat anti-Rabbit IgG H+L antibody were purchased from Abcam (Tokyo, Japan). Lactic acid and H_2_O_2_ was purchased from Wako (Osaka, Japan).

### 4.2. Bacterial Strains and 16S rRNA Sequencing

Two isolated strains were prepared in the following methods in this study. *L. plantarum* strain BF1-13 was isolated by the DSW laboratory of DHC Co. from the bag filter (pore size, 0.5 μm) at 800 m depth DSW in the pumping station in Izu-Akazawa, Shizuoka Prefecture and strain H-6 was isolated from a seaweed ‘*Polyopes* sp.’ collected in Okinawa Prefecture. The genomic DNA used in the analysis of 16S rRNA gene sequences was extracted by Achromopeptidase (Wako, Osaka, Japan). Extracted DNA was PCR amplified by TKs Gflex DNA Polymerase (Takara Bio, Shiga, Japan) and was sequenced on ABI PRISM 3130 × l Genetic Analyzer System by using BigDye Terminator v3.1 Cycle Sequencing Kit (Applied Biosystems, Waltham, MA, USA). The 16S rDNA sequences were assembled using ChromasPro 2.1 (Technelysium, South Brisbane, AUS). Phylogenetic analysis was performed by using ENKI software (TechnoSuruga Laboratory, Shizuoka, Japan). Strain JCM11125 as the standard strain was obtained from RIKEN BRC (Ibaraki, Japan).

### 4.3. Cell Culture

Caco-2 cells (No. RCB0988, RIKEN BRC, Japan) were cultured by HG-DMEM at 37 °C in 5% CO_2_. The cells at the density of 0.24 × 10^6^ cells/cm^2^ were seeded on each hanging cell culture insert (Millicell, *Φ* 6.5 mm, 1.0 μm pore size, PET; 0.3 cm^2^ area; Merck KGaA, Darmstadt, Germany) to construct the intestinal epithelial model. The medium was changed every two days until the monolayer was completed. The cells used were between passage 11 and 25.

### 4.4. Bacteria Incubation and the Preparation of Metabolites Containing CSs from the Isolated Strains

Both of the two strains and the standard strain were incubated in liquid MRS medium (pH 6.5) at 37 °C for 12 h and the CSs were separately obtained from the bacterial suspension (with final bacterial concentrations of strains BF1-13 and JCM11125 as 1.67 × 10^9^, 7.44 × 10^8^ CFU/mL, respectively) by centrifugation (13,200× *g*, 4 °C, 5 min) followed by the sterilization through 0.2 µm filter (ADVANTEC, Tokyo, Japan). For each CS, the ability of LA production by the strain was measured at 450 nm by a microplate reader (Model 550, Bio-Rad, Hercules, CA, USA) using Lactate Assay Kit-WST (DOJINDO, Kumamoto, Japan) after pH value was measured. The comparison of the growth temperature among three strains was investigated following the incubation at 5, 10, 20, 27, 37, 40, 45 °C, by using the MRS agar medium (pH 6.5) for 2 weeks. Salt tolerance was investigated by 0~10% (*m*/*v*) NaCl supplementation to the MRS agar medium (pH 6.5). Carbohydrate fermentation ability was conducted by using the nutrient-limited medium (pH 6.5) containing different substrates (0.4%, *m*/*v*) separately, including glucose, sorbitol, trehalose, xylose, arabinose, mannitol, and lactose. Salt tolerance, pH tolerance and carbohydrate fermentation ability of the strains were determined following the 2 weeks incubation at 37 °C.

### 4.5. Measurement of the Intestinal TJs Barrier Function

Intestinal epithelial barrier function was evaluated as the TEER value on the intestinal epithelial model using Caco-2 cells by Millicell-ERS-2 system (Millipore, Billerica, MA, USA). Caco-2 cells achieved to form the monolayer from 7 to 14 days after seeding when temperature corrected TEER (tcTEER) values of the model became stable around 200–300 Ω cm^2^ (310.15 K) [25]. In this study, the CSs of the two strains were prepared with HG-DMEM separately as the evaluation medium. Each evaluation medium was added to the apical side of the membrane and the model was cultured for 2 h (37 °C, 5% CO_2_) to be habituated. After culturing, all the media in both apical and basolateral sides were replaced by HG-DMEM without FBS due to its eliminative effect of H_2_O_2_. Then, HG-DMEM (without FBS) containing 1 mM H_2_O_2_ (negative control, NC) or without H_2_O_2_ (blank) was added to the basolateral side. The model was incubated for 6 h (37 °C, 5% CO_2_). TEER values were measured before and until 8 h of the CSs supplementation. Authentic LA (1.32, 1.97, and 2.63 mM) prepared by HG-DMEM was also administered to the apical side following the same procedure to investigate its effect on TJs barrier function.

### 4.6. Immunofluorescence Staining

Cell monolayer was cultured separately with various evaluation media, including 5% (*v/v*) CSs of the two strains or authentic LA (1.32 mM) for 2 h (37 °C, 5% CO_2_) to be habituated. After culturing, all the media were replaced by HG-DMEM without FBS. Then, the model was treated without (blank) or with 1 mM H_2_O_2_ (NC) for 2 h (37 °C, 5% CO_2_). After H_2_O_2_ treatment, the monolayers were rinsed with cold PBS, fixed in methanol at 4 °C for 30 min. Next, they were exposed to PBS containing 0.3% Triton-X100 for 5 min. Monolayers were blocked in PBS containing 5% (*v/v*) normal fetal serum for 30 min at room temperature. The monolayers were separately incubated with rabbit anti-CLDN-4, OCLN, AQP3 antibody overnight at 4 °C, followed by the 2nd antibody (horseradish peroxidase conjugated goat anti-rabbit IgG H+L antibody) for 1 h in the dark at room temperature. Each protein expression was observed by the fluorescence microscope with a magnification of 1000 (BX51N-33P, Olympus, Tokyo, Japan).

### 4.7. RNA Extraction and Quantitative RT-PCR

Confluent Caco-2 cells were preincubated without (blank and NC) or with 5% (*v/v*) CS of each strain or authentic LA (1.32 mM) for 2 h (37 °C, 5% CO_2_). After preincubation, all the media were replaced by HG-DMEM without FBS. Then, cells were treated with or without it (blank) 1 mM H_2_O_2_ for 0.5, 1, and 2 h, separately (37 °C, 5% CO_2_). After H_2_O_2_ treatment, total RNA was extracted using ISOGENII (NIPPON GENE, Tokyo, Japan) according to the supplier’s protocol by using sterile and RNase-free tubes. Total RNA was quantified by measuring optical density at 260 nm. cDNAs were synthesized using 1 μg total RNA by PrimeScriptTM Reverse Transcriptase (TaKaRa, Shiga, Japan) following the instruction. Quantitative RT-PCR was conducted with 2 µL cDNA template in a total volume of 20 µL by TB Green^®^ Fast qPCR Mix (TaKaRa, Shiga, Japan) using Applied Biosystems^®^ StepOnePlusTM Real-Time PCR Systems (Thermo Fisher, Waltham, MA, USA) with specific primers of each gene (Appendix A). PCR reaction for each sample was done in triplicate in 96-well plates. Gene expression was normalized to glyceraldehyde-3-phosphate dehydrogenase (GAPDH) and calculated by the comparative CT method to be expressed as fold-change compared to the blank.

### 4.8. Statistical Analysis

All the tcTEER values were indicated as relative TEER values by which the initial values were 100%. All values were indicated as mean ± SD. Statistical significance was determined by the U-Mann-Whitney test for the comparison between two groups and the Kruskal-Wallis test for that among groups.

## 5. Conclusions

In summary, metabolites produced by DSW-derived *Lactiplantibacillus plantarum* strain BF1-13 protect the intestinal epithelial barrier from the dysfunction caused by H_2_O_2_ treatment. It was also elucidated that the mechanism of this protective effect was achieved by both the enhancement of CLDN-4 expression and the suppression on AQP3-facililating H_2_O_2_ invasion. The oxidative stress induced by H_2_O_2_ impairs the intestinal epithelial barrier function which is mainly regulated by TJs, especially CLDN-4 [26]. The dysfunction of the intestinal epithelial barrier is correlated to the IBD, including ulcerative colitis (limited to the large intestine) and Crohn’s disease (mainly affects the small intestine), which correlates with a higher risk for colorectal cancer [27]. For the people who are at risk of IBD that could be triggered by both genetic susceptibility and environmental exposure [28], the daily intake of the food fermented by the strain BF1-13 could help them with maintaining the intestinal epithelial barrier function to lower the risk. For the patients who are suffering the prolonged inflammation, metabolites produced by the strain could be the supplements to support them in the daily diet or help them recover from the surgical treatment.

## Figures and Tables

**Figure 1 marinedrugs-20-00087-f001:**
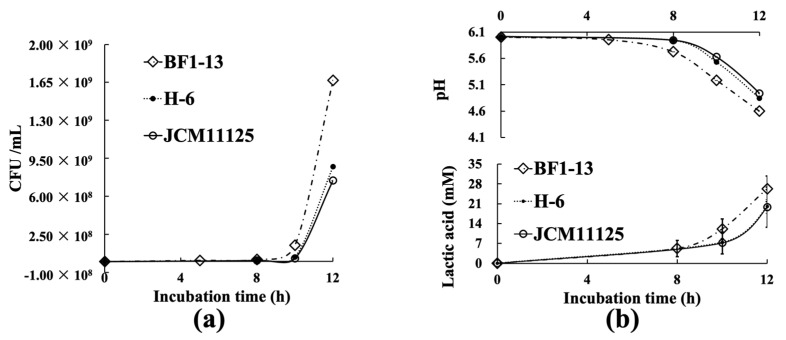
Comparison of each strain (BF1-13 ↓, H-6 λ, JCM11125 ϒ) on various characteristics. (**a**) The growth curve of the three strains. (**b**) The amounts of LA and pH changing of each strain during 12 h incubation. Data are shown as means ± SD, *n* = 9.

**Figure 2 marinedrugs-20-00087-f002:**
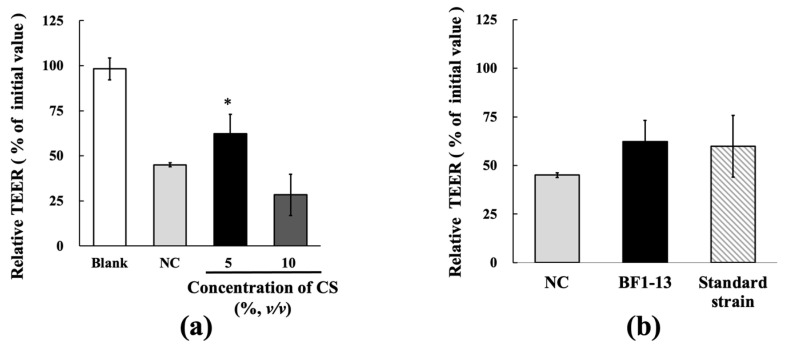
Effects of the metabolites of strain BF1-13 (black) and standard strain (slashes) on TJs barrier function in human Caco-2 cells. (**a**) The relationship between relative TEER value and CS supplemented concentration with 1 mM H_2_O_2_ (NC) or without (blank). (**b**) The comparison of the protective effect between the metabolites of the two strains on TJs barrier function. Monolayers were preincubated with 5% (*v/v*) CS of the two strains separately or without (NC) for 2 h, followed by 1 mM H_2_O_2_ treatment for 6 h or not (blank). Relative TEER values are means ± SD, *n* = 3. Asterisks indicate a significant difference with NC (* *p* < 0.05) determined by the Mann–Whitney U test.

**Figure 3 marinedrugs-20-00087-f003:**
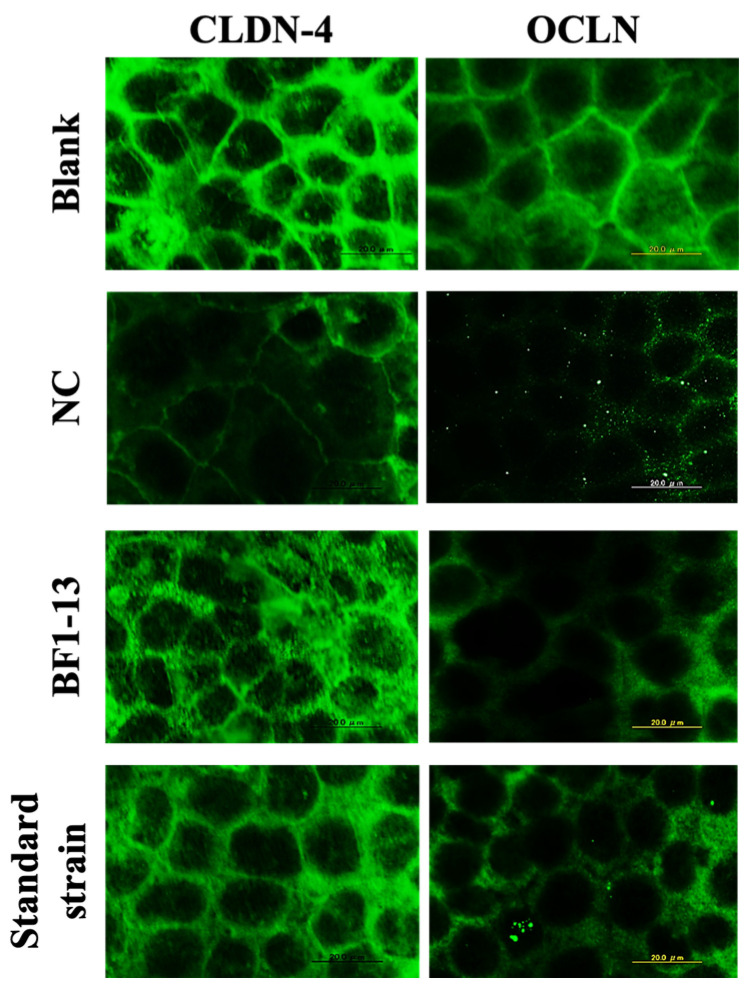
The comparison of the enhancements between the metabolites of the two strains on the expression of CLDN-4 and OCLN. Monolayers were preincubated with 5% (*v/v*) CS of the two strains separately or without (NC) for 2 h, followed by 1 mM H_2_O_2_ treatment for 2 h or not (blank). Images were observed by fluorescence microscopy. Each image is representative of 3 similar experiments. The scale bar is 20 µm.

**Figure 4 marinedrugs-20-00087-f004:**
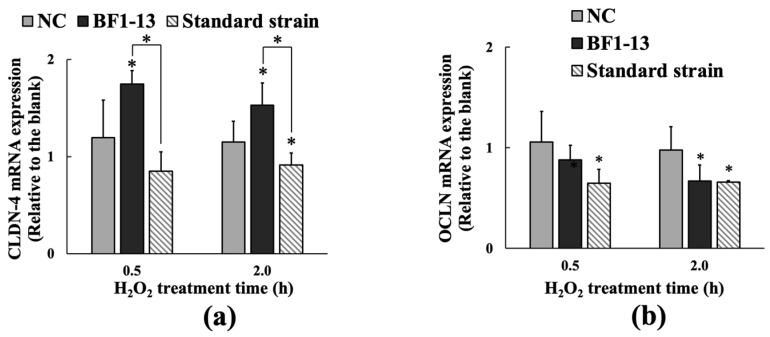
The inducement of the metabolites of the strain BF1-13 (black) on the expression of CLDN-4 (**a**) at the transcription level compared with the standard strain (slashes), which was compared with the expression of OCLN (**b**). Monolayers were preincubated with 5% (*v/v*) CS of the two strains separately or without (NC) for 2 h, followed by 1 mM H_2_O_2_ treatment for 0.5 and 2 h separately or not (blank). The mRNA expressions relative to the blank are shown as means ± SD, *n* = 3. Asterisks indicate a significant difference with the blank, or between BF1-13 and the standard strain (* *p* < 0.05) determined by the Mann–Whitney U test.

**Figure 5 marinedrugs-20-00087-f005:**
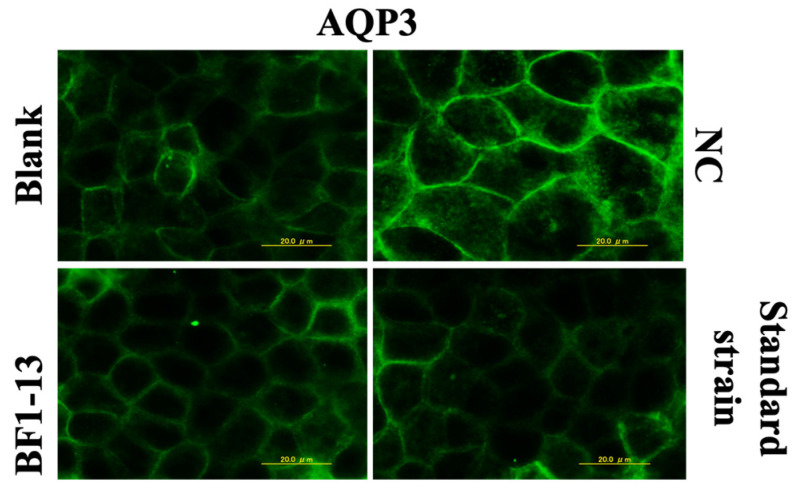
The suppressions of the AQP3 expression in human Caco-2 cells induced by the supplementation of the CSs. Monolayers were preincubated with 5% (*v/v*) CS of the two strains separately or without (NC) for 2 h, followed by 1 mM H_2_O_2_ treatment for 2 h or not (blank). Images were observed by fluorescence microscopy. Each image is representative of 3 similar experiments. The scale bar is 20 µm.

**Figure 6 marinedrugs-20-00087-f006:**
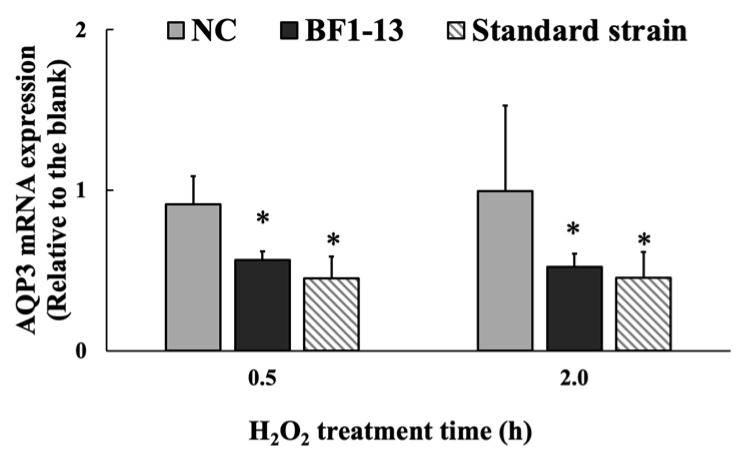
The suppressions of the metabolites of strain BF1-3 (black) and standard strain (slashes) on the expression of AQP3 at the transcription level. Monolayers were preincubated with 5% (*v/v*) CS of the two strains and LA (1.32 mM) separately or without (NC) for 2 h, followed by 1 mM H_2_O_2_ treatment for 0.5 and 2 h separately or not (blank). The mRNA expressions relative to the blank are shown as means ± SD, *n* = 3. Asterisks indicate a significant difference with the blank, or between BF1-13 and the standard strain (* *p* < 0.05) determined by the Mann–Whitney U test.

**Figure 7 marinedrugs-20-00087-f007:**
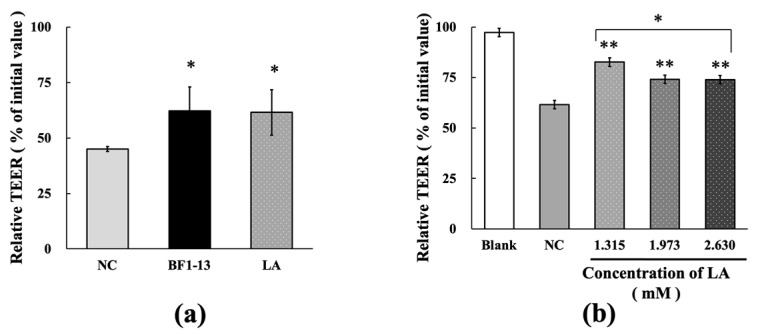
Protective effects of the metabolites of strain BF1-13 (black) and LA (spots) on TJs barrier function. (**a**) The comparison of the protective effect between 5% (*v/v*) CS and LA (1.32 mM) on TJs barrier function with 1 mM H_2_O_2_ (NC) or without (blank). (**b**) Dose-dependent protective effect of authentic LA on TJs barrier function. Monolayers were pre-incubated with 1.32, 1.97, and 2.63 mM authentic LA separately or without (NC) for 2 h, followed by 1 mM H_2_O_2_ treatment for 6 h or not (blank). Relative TEER values are means ± SD, *n* = 3 (**a**), 6 (**b**). Asterisks indicate a significant difference with NC (* *p* < 0.05, ** *p* < 0.005) determined by the Mann–Whitney U test, or among various concentrations of LA (* *p* < 0.01) determined by the Kruskal–Wallis test.

**Figure 8 marinedrugs-20-00087-f008:**
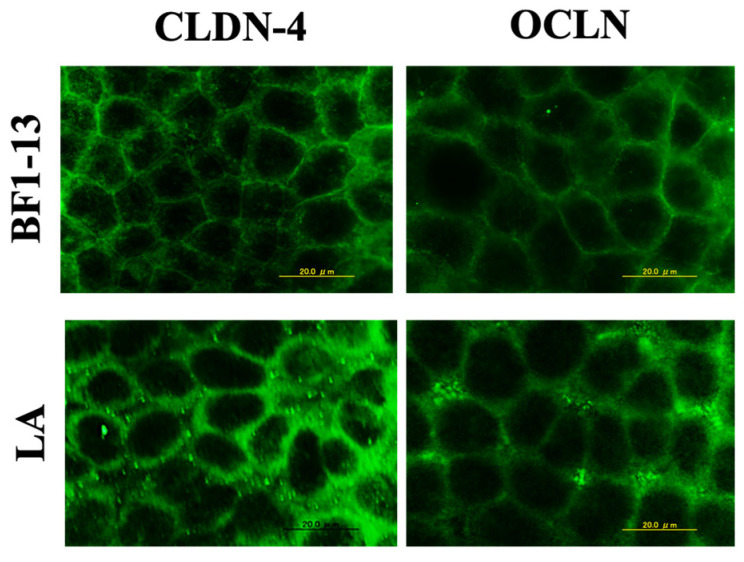
The comparison of the enhancements between the metabolites of strain BF1-13 and LA on the expression of CLDN-4 and OCLN. Monolayers were preincubated with 5% (*v/v*) CS of the strain and LA (1.32 mM) separately for 2 h, followed by 1 mM H_2_O_2_ treatment for 2 h. Images were observed by fluorescence microscopy. Each image is representative of 3 similar experiments. The scale bar is 20 µm.

**Figure 9 marinedrugs-20-00087-f009:**
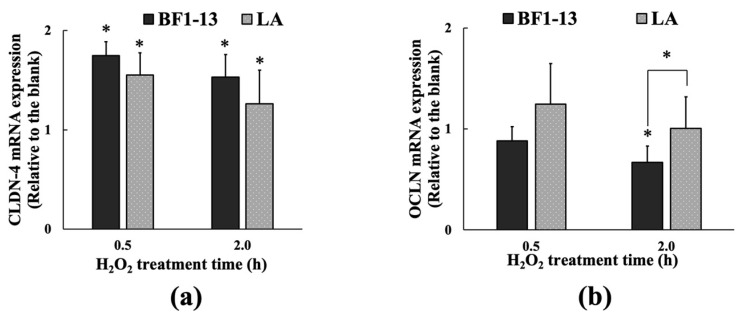
The comparison of the effect between the metabolites of strain BF1-13 (black) and LA (spots) on the expression of CLDN-4 (**a**) and OCLN (**b**) at the transcription level. Monolayers were preincubated with 5% (*v/v*) CS of strain BF1-13 and LA (1.32 mM) separately or without (NC) for 2 h, followed by 1 mM H_2_O_2_ treatment for 0.5 and 2 h separately or not (blank). The mRNA expressions relative to the blank are shown as means ± SD, *n* = 3. Asterisks indicate a significant difference with the blank, or between BF1-13 and LA (* *p* < 0.05) determined by the U-Mann-Whitney test.

**Figure 10 marinedrugs-20-00087-f010:**
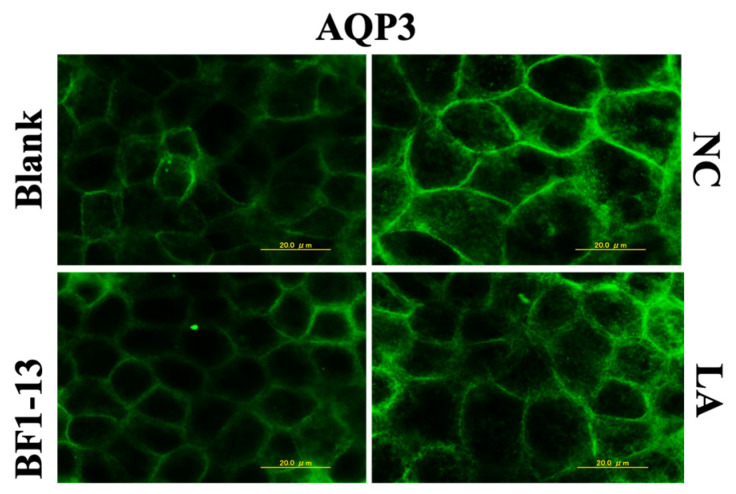
The effect of the metabolites of strain BF1-13 and equal concentration of authentic LA on the expression of APQ3 in human Caco-2 cells. Monolayers were preincubated with 5% (*v/v*) CS of the strain BF1-13 and LA (1.32 mM) separately or without (NC) for 2 h, followed by 1 mM H_2_O_2_ treatment for 2 h or not (blank). Images were observed by fluorescence microscopy. Each image is representative of 3 similar experiments. The scale bar is 20 µm.

**Figure 11 marinedrugs-20-00087-f011:**
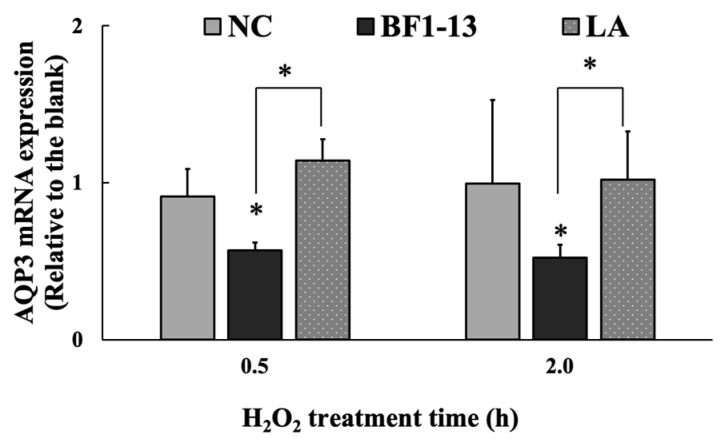
The effect of the metabolites of strain BF1-3 (black) and equal concentration of authentic LA (spots) on the expression of AQP3 at the transcription level. Monolayers were preincubated with 5% (*v/v*) CS of the strain BF1-13 and LA (1.32 mM) separately or without (NC) for 2 h, followed by 1 mM H_2_O_2_ treatment for 0.5 and 2 h separately or not (blank). The mRNA expressions relative to the blank are shown as means ± SD, *n* = 3. Asterisks indicate a significant difference with the blank, or between BF1-13 and LA (* *p* < 0.05) determined by the U-Mann-Whitney test.

**Table 1 marinedrugs-20-00087-t001:** Comparison of growth characteristics of the three strains.

		Strain BF1-13	Strain JCM11125	Strain H-6
Temperature (°C)	5	−	−	−
	10	−	−	−
	20	−	+	+
	27	+	+	+
	37	+	+	+
	40	+	+	+
	45	−	−	−
pH	2	+	+	-
	3	+	+	+
	4	+	+	+
	6.5	+	+	+
	9	+	+	+
	10	−	−	−
	11	−	−	−
NaCl (%, *m/v*)	0	+	+	+
	1	+	+	+
	2	+	+	+
	3	+	+	+
	5	+	+	+
	10	−	−	−
Fermentation substrate	Glucose	+	+	+
	Sorbitol	+	+	+
	Trehalose	+	+	+
	Xylose	+	+	+
	Arabinose	+	+	+
	Mannitol	+	+	+
	Lactose	+	+	+

+, growth; -, no growth (*n* = 3).

**Table 2 marinedrugs-20-00087-t002:** Characteristics of each strain after 12 h incubation.

Strain	BF1-13	JCM11125	H-6
CFU/mL	1.67 × 10^9^	7.44 × 10^8^	8.71 × 10^8^
pH	4.6	4.9	4.9
Latic acid (mM)	26.3	19.8	20.0

The number of bacterial cells indicates as CFU/mL (*n* = 9, mean ± SD).

## Data Availability

The data that support the administration concentration of hydrogen peroxide used in this study are openly available at http://doi.org/10.2741/3223 (accessed on 1 May 2008).

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
