# Peer review of "Metabolites Produced by a New Lactiplantibacillus plantarum Strain BF1-13 Isolated from Deep Seawater of Izu-Akazawa Protect the Intestinal Epithelial Barrier from the Dysfunction Induced by Hydrogen Peroxide"

_marinedrugs, 2022, doi:10.3390/md20020087_

Round 1

Reviewer 1 Report

The manuscript presents a new, complex and promising research on the protection given by the  metabolites produced by a strain of lactic acid bacteria isolated from deep seawater against dysfunction caused by H2O2 treatment in the intestinal epithelial barrier. The authors conducted a comparative study of three strains of lactic acid bacteria (strain BF1-13, strain K-6 and the standard strain JCM11125) after which they chose the L. plantarum strain which has the fastest growth and the highest lactic acid production. The chosen methods demonstrated that the protection is due to both the CLDN-4 expression enhancement and the suppression on AQP3-facililating H2O2 invasion.

The authors point out that it is the first time to report LA to be an essential substance for the enhancement on the intestinal epithelial barrier from the dysfunction caused by H2O2. However, the presence of other primary or secondary metabolites in CS obtained from the culture of BF1-13 bacteria, which could influence the protective mechanism, should also be taken into account. Thus, I consider that a more detailed analysis of the properties of the L. plantarum strain used in this research is necessary or, if its biochemical characteristics are known, it should be mentioned briefly in the text.

Observations:

- The results of the experiments are a little difficult to follow due to the numerous abbreviations

- Unexplained abbreviations are used in some places in the text, for example CS - probably supernatant culture or SSW (line 272)

- It would be better to avoid sentences that start with the conjunction "And"

- Line 329 - H2O2 -

- Check which words should be bolded in Table 1

- What is the reason for choosing the value of 5% (v / v) CSs of the bacterial strains?

Author Response

List of Actions

LOA1: All the sentences which started with “And” have been revised. (line 37, 128, 146, 214, 382)

LOA2: The abbreviation “ROS” has been corrected back to the fully-spelling “reactive oxygen species”. (line 47, 246)

LOA3: The origin where strain JCM11125 was isolated and the accession number have been added. (line 77-79)

LOA4: Comparisons of salt tolerance, pH tolerance and range of growth temperature have been added to the Results. (line 85-88 & Table 1.)

LOA5: Table 1. was changed to Table 2. and the mistakenly bolded words have been corrected

LOA6: Strain “K-6” has been corrected as the formally registered name (in DDBJ) “H-6” after the discussion with all the co-authors. (Table 1, 2. & line 97)

LOA7: The unexplained abbreviation “CS” has been explained as “culture supernatant” at its first appearance (line 95), “SSW” has been corrected as “surface seawater”. (line 279)

LOA8: 4.2 Method of identification of examined strains (H-6 & BF1-13) has been clarified in the Materials and Methods. (line 303-310)

LOA9: Conditions for the comparisons of salt tolerance, pH tolerance, and range of growth temperature have been clarified in the Materials and Methods. (line 326-330)

LOA10: “H2O2” has been corrected to “H2O2”. (line 350)

Dear reviewer:

-Thank you for your considerate thought concerning the abbreviations used in this paper. As you suggested, except for the necessary ones, we deleted some of them, see LOA2 & 7.

-The expression of the sentences starting with “And” has been improved, see LOA1.

-Format mistakes have been corrected, see LOA5 & 10.

-Thank you for your question about the concentration of CSs which was supplemented. We have conducted both the TEER assay and immunofluorescent microscopy with 5% and 10% CS of strain BF1-13. According to the TEER assay, the inhibition on the decreasing TEER values showed by 5% disappeared in the case of 10% CS. Also, there was no difference between 5% and 10% CS observed by immunofluorescent microscopy. Because this paper mainly focused on the new effect on TJs and AQP3-facilitating H2O2 transportation, 5% of CS was chosen for further experiments in this study. We may conduct further experiments especially related to the concentrations of the CS in the future as you kindly mentioned.

-As you suggested, more comparisons of characteristics among the strains including salt tolerance, pH tolerance, and range of growth temperature have been added, see LOA4 & 9.

Reviewer 2 Report

The manuscript by Diao et al. reports about the protective effect of the metabolites produced by a new Lactiplantibacillus plantarum strain BF1-13 isolated from deep seawater (DSW) on the intestinal epithelial barrier. Unfortunately, some necessary data are missing in the text of the manuscript. The strains used in the research are not described in the scientific literature. Moreover, there are no data in the text of manuscript regarding experimental and standard bacterial strains used, except the information that “two isolated strains L. plantarum strain BF1-13 was isolated by the DSW laboratory of DHC Co. … and strain K-6 was isolated from a seaweed ‘Polyopes sp.’ collected in Okinawa Prefecture. And strain JCM11125 as the standard strain was obtained from RIKEN BRC 302 (Japan).”

Whole genome sequencing of examined bacterial strains must be performed and deposited in a publicly available database, and the relevant accession numbers must be provided in the text of manuscript. These data are also necessary in order to confirm identification of isolated bacterial strains. Namely, there are no data in the text of manuscript about method of identification of examined strains. These data must be provided in order to consider if manuscript is appropriate for publication.

Additionally, several other improvements are needed in terms of clarity.  Moreover, English must be improved.  Therefore, the submitted manuscript, in its current form, does not achieve the standard necessary for the publication in Marine Drugs journal.

Author Response

List of Actions

LOA1: All the sentences which started with “And” have been revised. (line 37, 128, 146, 214, 382)

LOA2: The abbreviation “ROS” has been corrected back to the fully-spelling “reactive oxygen species”. (line 47, 246)

LOA3: The origin where strain JCM11125 was isolated and the accession number have been added. (line 77-79)

LOA4: Comparisons of salt tolerance, pH tolerance, and range of growth temperature have been added to the Results. (line 85-88 & Table 1.)

LOA5: Table 1. was changed to Table 2. and the mistakenly bolded words have been corrected

LOA6: Strain “K-6” has been corrected as the formally registered name (in DDBJ) “H-6” after the discussion with all the co-authors. (Table 1, 2. & line 97)

LOA7: The unexplained abbreviation “CS” has been explained as “culture supernatant” at its first appearance (line 95), “SSW” has been corrected as “surface seawater”. (line 279)

LOA8: 4.2 Method of identification of examined strains (H-6 & BF1-13) has been clarified in the Materials and Methods. (line 303-310)

LOA9: Conditions for the comparisons of salt tolerance, pH tolerance, and range of growth temperature have been clarified in the Materials and Methods. (line 326-330)

LOA10: “H2O2” has been corrected to “H2O2”. (line 350) 

Dear reviewer:

Thank you for your objective and precise opinion on the information about the strains used in this study.

-The accession number of the standard strain JCM11125 has been added and more information about its isolated origin has been clarified, see LOA3.

-The method of 16S rRNA sequencing of strain H-6 (corrected from K-6, see LOA6) and BF1-13 has been clarified, see LOA8.

-I just received the DDBJ (DNA Data Bank of Japan) accession number of the genomic sequencing of the two strains: Strain H-6 (LC666820) and Strain BF1-13 (LC666821).

According to DDBJ, the data would be released online to the public by Friday.

Round 2

Reviewer 2 Report

General comment:

Authors mentioned and examined only lactic acid, and no other metabolites of examined strain Lactiplantibacillus plantarum. Therefore, it is necessary to be precise and if there are any other metabolite which was recognized to have an effect, please specified or avoid word 'metabolites' and use only precise metabolite which was detected and examined. It must be corrected through whole text of manuscript.

Additionally, several other improvements are needed in terms of clarity.  Moreover, English must be improved.  Therefore, the submitted manuscript, in its current form, does not achieve the standard necessary for the publication in Marine Drugs journal.

Specific comments:

Title

- Must be corrected because it was shown that only lactic acid, and no other produced metabolites, have an impact on intestinal epithelial barrier

- Origin of examined strain can also be excluded from the title because the metabolite shown to have an effect is common metabolite of all strains of the same species Lactiplantibacillus plantarum

Introduction

The aim of study must be described at the end of this chapter!

Line 74. – word Lactobacillus must be italic! (must be also corrected in the chapter References no. 18-22)

Line 82. – 'Especially, it can only convert 80% of the glucose which can be used as a good comparison with strain BF1-13 on the investigation of LA [16].' -  The sentence is not clear. Re-formulate and rather introduce into the chapter Discussion when compared with the results obtained by examined BF-13 strain.

Results

 At the beginning of the first chapter 2.1. Characteristics of Three Lactiplantibacillus Strains, the method of identification of isolated strains L. plantarum strain BF1-13 and K-6 must be mentioned.

In all figures and tables

At least the labels of the tested bacterial strains must be mentioned in the titles of figures.

Figure 1.

- there are no data shown on the graph regarding lactic acid concentration! The authors must express lactate or lactic acid, and correct on the same way through the text of manuscript

Figures 3 and 4.

There are shown results for strain BF1-13 and control strain. Please correct in the titles of the figures.

Figure 11.

- replace ‘The effect by’ with ‘The effect of’

Table 1.

Salt tolerance (%) - percent must be expressed as % (by volume) or % (m/V).

Temperature range (℃) – must be defined (T range of growth?!)

Table 2.

- Replace ‘LA accumulation’ with ‘Lactic acid’.

  1. Materials and Methods

Lines 299-302 – introduce accession numbers

Author Response

Dear reviewer:

Thank you for your careful review of this paper.

- The aim of this study has been clarified in the last part of the Introduction. See LOA3

- Word “Lactobacillus” has been corrected to be italic. see LOA2.

- The evidence which supports the difference of lactic acid production between strain BF1-13 and standard strain has been reformulated and clarified. See LOA15, 24.

- The method of identification and accession numbers of isolated strains have been added to the Results 2.1. See LOA4.

- “Lactate” has been corrected in the same way as “lactic acid” except the lactate assay kit commercial name. See LOA9.

- The labels of the tested strains and LA have been added in the titles of all figures. See LOA10.

- The specific effects have been clarified in the titles of the figures with shown results. See LOA11.

- “The effect by” has been replaced with “The effect of”. See LOA13.

- Salt tolerance has been expressed as “% (m/v)”. See LOA19.

- The description of growth temperature has been modified. See LOA 7 & 18.

-The DDBJ (DNA Data Bank of Japan) accession number of strain BF1-13 and H-6 has been added. See LOA4.

Thank you for your opinion on which was bioactive compound contained in the metabolites produced by strain BF1-13. According to this, may I have some explanations according to the results of this study?

Before stating the explanation, we want to mention that MRS medium which was used for the bacterial incubation showed no effect on the intestinal epithelial barrier according to TEER assay. (data could be provided if needed)

First, the comparison between the CS of strain BF1-13 and lactic acid (LA):

1-1. Nearly the same relative TEER values showed by the supplementation of CS of BF1-13 and an equal amount of LA, which indicated the same protection on the intestinal epithelial barrier function can be achieved by an equal amount of LA.

1-2. Both CS and LA induced CLDN-4 expression according to immunofluorescent staining and RT-PCR. However, CS showed higher inducement on mRNA expression than LA, which indicated some other metabolites also enhance CLDN-4.

1-3. LA showed more enhancement on OCLN (significantly differed from BF1-13 on mRNA expression at 2h) according to immunofluorescent staining and RT-PCR, which indicated LA does enhance OCLN as well.

1-4. Only CS suppressed AQP3 expression, which was significantly different from LA.

Then, the comparison between the CS of strain BF1-13 and standard strain:

2-1. BF1-13 showed a slightly higher relative TEER value than the standard strain., which was caused by the higher concentration of LA contained in the CS.

2-2. BF1-13 induced CLDN-4 expression (significantly differed from the standard strain) according to immunofluorescent staining and RT-PCR, which indicated some other compounds only existed in the metabolites produced by strain BF1-13. (combining 1-2)

2-3. BF1-13 showed higher mRNA expression of OCLN, which was caused by the higher concentration of LA contained in the CS. (combining 1-3)

2-4. Both the CSs of the two strains suppressed AQP3 expression with no significant difference, which indicated some common compounds in the metabolites produced by Lactiplantibacillus sp.

In summary, LA contained in the metabolites does protect the intestinal epithelial barrier function through the enhancement of the TJs expression.

As you suggested, some common compounds in the metabolites produced by Lactiplantibacillus sp. were responsible for the suppression of AQP3.

However, there exist some specific compounds only existed in the metabolites produced by strain BF1-13 which can also enhance CLDN-4 expression.

In this study, the protective effect of the metabolites produced by strain BF1-13 also LA on the intestinal epithelial barrier has been indicated. It provides the potential application of them as supplements for the patients who are suffering IBD. In this case, the “metabolites” and strain-specific part in the title should be kept in our opinion. Indeed, further study on the specific compound investigated by analytic chemistry and the mechanism underlying the effect investigated by animal experiments are to be conducted.

Round 3

Reviewer 2 Report

The authors accepted all the requested remarks and answered all issues, what contributed to manuscript improvement. Therefore, the manuscript is appropriate for publication.

Author Response

Dear reviewer:

Thank you for your precious suggestions for this paper and your acknowledgment of the improvements in the revised version.